# A binary pulsar in a 53-minute orbit

Z. Pan[1,2,3,4,16], J. G. Lu[1,2,3,4,16], P. Jiang[1,2,3,4 ✉], J. L. Han[1,3,4 ✉], H.-L. Chen[5,16], Z. W. Han[5,6], K. Liu[7], L. Qian[1,2,3,4], R. X. Xu[8,9,10], B. Zhang[11,12 ✉], J. T. Luo[13], Z. Yan[3,4,14], Z. L. Yang[1,3,4], D. J. Zhou[1,3,4], P. F. Wang[1,3,4], C. Wang[1,3,4], M. H. Li[15] & M. Zhu[1,2,3,4]

Spider pulsars are neutron stars that have a companion star in a close orbit. The companion star sheds material to the neutron star, spinning it up to millisecond rotation periods, while the orbit shortens to hours. The companion is eventually ablated and destroyed by the pulsar wind and radiation[1,2]. Spider pulsars are key for studying the evolutionary link between accreting X-ray pulsars and isolated millisecond pulsars, pulsar irradiation effects and the birth of massive neutron stars[3–6]. Black widow pulsars in extremely compact orbits (as short as 62 minutes[7]) have companions with masses much smaller than $0.1\,M_\odot$. They may have evolved from redback pulsars with companion masses of about $0.1$–$0.4\,M_\odot$ and orbital periods of less than 1 day[8]. If this is true, then there should be a population of millisecond pulsars with moderate-mass companions and very short orbital periods[9], but, hitherto, no such system was known. Here we report radio observations of the binary millisecond pulsar PSR J1953+1844 (M71E) that show it to have an orbital period of 53.3 minutes and a companion with a mass of around $0.07\,M_\odot$. It is a faint X-ray source and located 2.5 arcminutes from the centre of the globular cluster M71.

PSR J1953+1844 (M71E) was discovered in 2021 using the Five-hundred-meter Aperture Spherical radio Telescope (FAST)[10,11] by the FAST Galactic Plane Pulsar Snapshot (GPPS) survey and identified as a binary[12]. Using the archival data from the FAST Globular Cluster Pulsar survey[13], its orbital parameters were determined. Follow-up observations for timing started in September 2021. Its spin, astrometric and orbital parameters have been measured by means of pulsar timing (Table 1). The residuals and polarization pulse profiles can be found in Methods. Its orbital period is 53.3 minutes; the previous records of binary pulsar orbital periods were 75 minutes from the binary gamma-ray pulsar J1653-0158 (ref. 14) and 62 minutes from the optical source ZTF J1406+1222 (ref. 7). The dispersion measure value of the pulsar is $113.1\,\mathrm{cm^{-3}\,pc}$, which is close to, but lower than, these of the other four known pulsars from M71 (M71A, $117\,\mathrm{cm^{-3}\,pc}$ (ref. 15), and M71B–D, around $116$–$119\,\mathrm{cm^{-3}\,pc}$ (ref. 13)). Its timing position (J2000 19:53:37.95 +18:44:54.3) is approximately 2.5 arcminutes away from the centre of M71 (J2000 19:53:46.49 +18:46:45.1 (refs. 16,17)). A Chandra X-ray counterpart was identified almost at the timing position, whereas no optical (Sloan Digital Sky Survey (SDSS)), infrared (Two Micron All-Sky Survey (2MASS)) or gamma-ray (Fermi satellite) counterpart was identified around the timing position from the archival data. The details of the Chandra data can be found in Methods. The core radius, projected half-light radius and half-mass radius (three-dimensional) of M71 are 1.07, 3.32 and 4.81 pc (refs. 18,19), respectively. (Note that the updated numbers from ref. 19 were used.) Assuming the distances to the pulsar and the globular

cluster are the same (4 kpc; ref. 20), this pulsar is not in the core region, but still in the outskirts, of M71. Therefore, it is debatable whether this pulsar is a member of this globular cluster. We name it as M71E because of its proximity to M71 and it is very probably a binary in M71. Assuming an edge-on orbit and a pulsar mass of $1.4\,M_\odot$ (the typical mass of a neutron star), and with the timing solution, the minimum companion mass was derived to be $0.008\,M_\odot$. It is worth noting that no eclipsing events were observed in all the observations, indicating that the orbit is not edge-on and the companion mass should be larger than the minimum.

As the mass distribution of millisecond pulsars can be bimodal and the mass of pulsars in spiders tends to be heavier[5,6], the possible mass of the neutron star in M71E was assumed to be 1.0, 1.4 or $2.0\,M_\odot$ for additional analyses. The tight orbit eliminates the possibility of a main-sequence companion star, with only white dwarfs or brown dwarfs being dense enough to reach such a period. On the basis of the mass function derived from M71E timing, one can pose a generic constraint on the orbital inclination angle versus companion mass (Fig. 1).

We first consider a white dwarf companion. The observational lower limit of white dwarf masses in a detached binary system is about $0.16\,M_\odot$ (ref. 21) and the theoretical mass lower limit is about $0.14\,M_\odot$, below which white dwarfs are hard to form from binary evolution[22]. As shown in Fig. 1a, for a wide range of assumed pulsar masses, to have a companion mass of $0.16\,M_\odot$ or higher, the allowed inclination angle should be very face-on, that is, $i < 3.6°$. For a random distribution of orbital inclination angle, the probability of having such a small

[1]National Astronomical Observatories, Chinese Academy of Sciences, Beijing, People's Republic of China. [2]Guizhou Radio Astronomical Observatory, Guizhou University, Guiyang, People's Republic of China. [3]College of Astronomy and Space Sciences, University of Chinese Academy of Sciences, Chinese Academy of Sciences, Beijing, People's Republic of China. [4]Key Laboratory of Radio Astronomy and Technology, Chinese Academy of Sciences, Beijing, People's Republic of China. [5]Yunnan Observatories, Chinese Academy of Sciences, Kunming, People's Republic of China. [6]University of Chinese Academy of Sciences, Chinese Academy of Sciences, Beijing, People's Republic of China. [7]Max-Planck-Institut für Radioastronomie, Bonn, Germany. [8]Department of Astronomy, Peking University, Beijing, People's Republic of China. [9]Kavli Institute for Astronomy and Astrophysics, Peking University, Beijing, People's Republic of China. [10]State Key Laboratory of Nuclear Physics and Technology, School of Physics, Peking University, Beijing, People's Republic of China. [11]Nevada Center for Astrophysics, University of Nevada, Las Vegas, NV, USA. [12]Department of Physics and Astronomy, University of Nevada, Las Vegas, NV, USA. [13]National Time Service Center, Chinese Academy of Sciences, Xi'an, China. [14]Shanghai Astronomical Observatory, Chinese Academy of Sciences, Shanghai, People's Republic of China. [15]State Key Laboratory of Public Big Data, Guizhou University, Guiyang, People's Republic of China. [16]These authors contributed equally: Z. Pan, J. G. Lu, H.-L. Chen. ✉e-mail: pjiang@bao.ac.cn; hjl@bao.ac.cn; bing.zhang@unlv.edu

## Table 1 | Timing parameters for M71E

| Pulsar name | M71E/J1953+1844 |
| --- | --- |
| MJD range | 59293-59781 |
| Data span (days) | 488 |
| Number of times of arrival | 635 |
| Timing residuals root mean square (r.m.s.) (μs) | 35.138 |
| **Measured quantities** | |
| Right ascension, RA (J2000) | 19:53:37.9464(1) |
| Declination, dec. (J2000) | +18:44:54.310(2) |
| Spin frequency (Hz) | 225.01840471114(7) |
| Spin frequency derivative (s$^{-2}$) | $-1.11(1)\times10^{-15}$ |
| Orbital period, $P_b$ (days) | 0.0370398638(5) |
| Time of ascending node, TASC (MJD) | 58829.26006(1) |
| Projected semimajor axis, $\chi_p$ (lt-s) | 0.006668(2) |
| $\hat{x}$ component of the eccentricity, $\kappa$ | 0.0003(6) |
| $\hat{y}$ component of the eccentricity, $\eta$ | 0.0005(6) |
| **Set quantities** | |
| Reference epoch (MJD) | 59474.630459 |
| Dispersion measure (cm$^{-3}$ pc) | 113.1 |
| Solar System ephemeris | DE440 |
| Binary model | ELL1 |

The ELL1 binary timing model[33] was chosen as the eccentricity of the orbit turns out to be small. The mass function obtained from the timing solution is 2.3×10$^{-7}$ $M_\odot$.

inclination angle is less than 0.3%, even if the pulsar mass is 2.0 $M_\odot$. Furthermore, there is no immediate binary evolution channel to form a detached white dwarf–neutron star binary with such a tight orbit. We therefore disfavour a white dwarf as the companion of M71E.

In view of the prevalence of spider pulsars in the sky, a more natural companion type would be a stripped dwarf star whose mass has been significantly depleted by mass transfer and the strong pulsar wind. The mass of such a companion can be constrained by requiring that the companion is strictly confined inside its Roche lobe. Otherwise, Roche lobe overflow would have quenched the coherent radio emission of the pulsar. In Fig. 1b, taking the typical pulsar mass of 1.4 $M_\odot$, we plot the mass–radius relationships of brown dwarf (BD)/low-mass stars for solar (Low-Mass star X (LMX) model) and zero (Low-Mass star Z (LMZ) model) metallicities[23], overlapped with the Roche lobe radius as a function of mass, which serves as the upper limit. One can then identify the dotted regions as the allowed parameter space. The constraint on the companion mass and corresponding density by the Roche lobe limit also meets other estimates for compact binaries[24]. For the pulsar mass from 1.0 to 2.0 $M_\odot$, the allowed companion mass is in the range 0.047–0.097 $M_\odot$, with a typical value of around 0.07 $M_\odot$. Such a mass range still requires a nearly face-on system, with the inclination angle in the range of 3.8°–12.1°, with a typical value of about 8°. Such a face-on geometry is also supported by the fact that no eclipsing was observed across the entire orbit, in contrast to most spider pulsars, which often show eclipsing features. The orbital inclination angle and companion mass from different model parameters are presented in Table 2.

The observation of such a short-period binary neutron star can be used to place constraints on binary evolution models. Figure 2 shows the companion masses (greater than 0.005 $M_\odot$) and orbital periods (less than 1 day) of all known short-orbital period binary pulsars regardless of whether their companions are white dwarfs or brown dwarfs, with the evolutionary tracks above these binaries. The horizontal bars indicate the estimated companion masses with the 90% probability when the pulsar mass is assumed to be 1.4 $M_\odot$. The black widows are distributed in the left part of the figure and the redbacks are in the right. M71E is denoted as the red star at the bottom. One can see that it

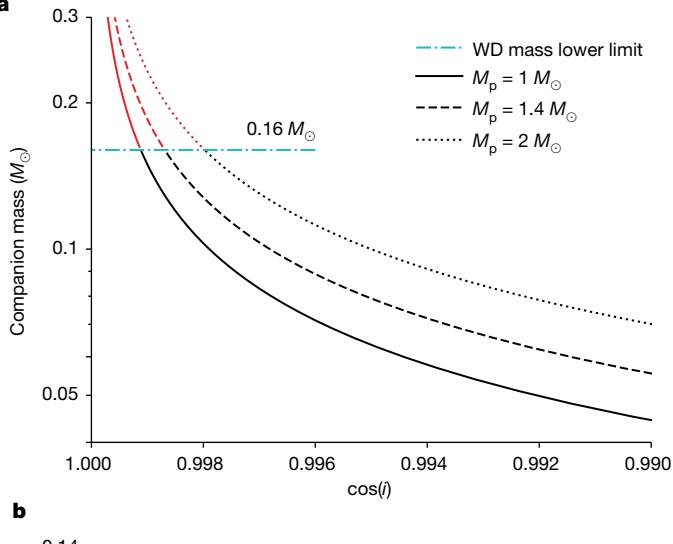

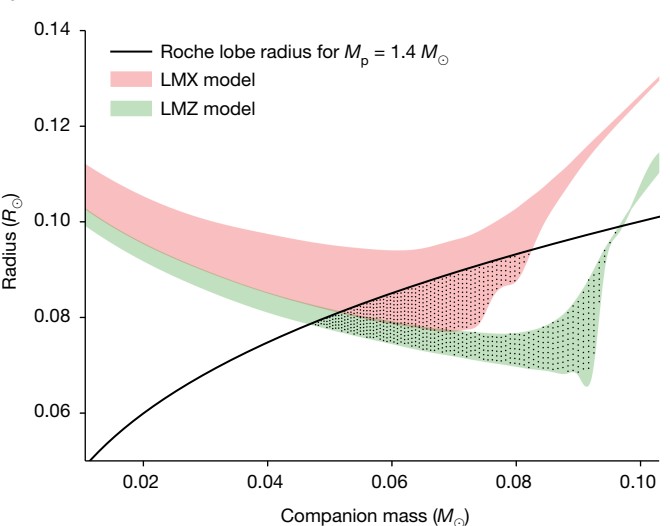

**Fig. 1 | Constraints on the mass and radius of the companion star. a**, A generic constraint on the companion mass as a function of inclination angle, *i*, on the basis of the measured mass function for different pulsar masses of 1.0, 1.4 and 2.0 $M_\odot$. If the orbital plane is randomly distributed, the integrated probability of inclination angle *i* from 90° (edge-on) to 0° (face-on) can be calculated as cos(*i*). The observed lower limit of a white dwarf (WD) mass is 0.16 $M_\odot$ (light-blue dashed–dotted horizontal line). The probability of the companion being a white dwarf is smaller than 0.3%. **b**, Constraints on the mass and radius of the companion star. The red and green regions denote the mass–radius relations of the companion star at solar (LMX model) and zero (LMZ model) metallicities[23], respectively, with an evolution age of between 3 and 10 Gyr. Assuming a pulsar mass of 1.4 $M_\odot$ ($M_p$), the radius of the Roche lobe as a function of companion mass is plotted as the black curve, which serves as the upper limit of the allowed parameter space. The dotted regions are the allowed regions for the companion mass and radius.

pushes the orbital period to a much tighter regime, but with a moderate companion mass, which is located between the typical companion masses of black widows and redbacks.

The formation channels of spider pulsars are still subject to debate[8,9,25,26]. Given that this system may reside in the globular cluster, it is, in principle, possible that it is the product of an exchange interaction[27,28]. However, because of the lack of a self-consistent calculation for this channel, it is unclear whether the specific parameters of M71E can be explained with the dynamical channel. In addition, we cannot exclude the possibility that this system is produced from the evolution of a neutron star–He/CO white dwarf system with a steady mass

**Table 2 | Modelled orbital inclination angles and companion masses**

| Pulsar mass ($M_\odot$) | Model | Evolutionary age (Gyr) | Range of inclination angle (degree) | Range of companion mass ($10^{-2} M_\odot$) |
|---|---|---|---|---|
| 1 | LMX | 0.6 | — | — |
| 1 | LMX | 10 | 6.9–4.5 | 5.3–8.1 |
| 1 | LMZ | 0.6 | 6.9–3.9 | 5.3–9.5 |
| 1 | LMZ | 10 | 7.6–3.8 | 4.8–9.7 |
| 1.4 | LMX | 0.6 | — | — |
| 1.4 | LMX | 10 | 8.7–5.5 | 5.2–8.2 |
| 1.4 | LMZ | 0.6 | 8.8–4.7 | 5.2–9.6 |
| 1.4 | LMZ | 10 | 9.5–4.7 | 4.7–9.7 |
| 2 | LMX | 0.6 | — | — |
| 2 | LMX | 10 | 11.1–7.0 | 5.1–8.2 |
| 2 | LMZ | 0.6 | 11.1–6.0 | 5.1–9.6 |
| 2 | LMZ | 10 | 12.1–6.0 | 4.7–9.7 |

The mass–radius relations of two typical models of low-mass stars are considered: LMX (metallicity $Z$ = metallicity of the Sun $Z_\odot$ and helium abundance $Y$ = 0.25) and LMZ (metallicity $Z$ = 0 and helium abundance $Y$ = 0.25)[23].

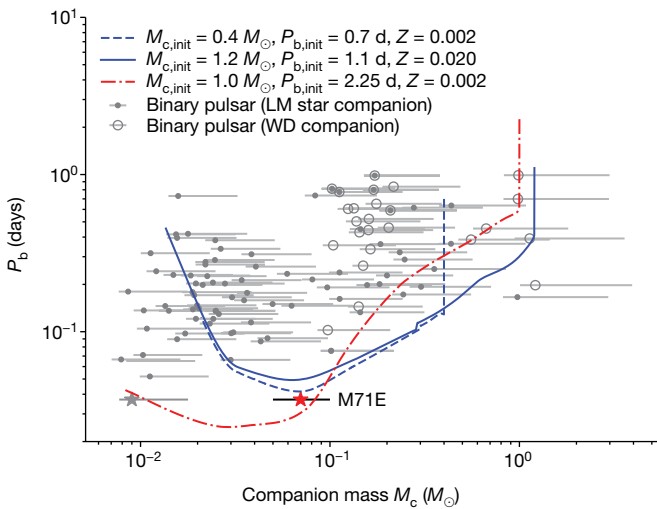

**Fig. 2 | Location of M71E on the companion mass versus orbital period plane and comparison with binary evolutionary models.** Known short-period binary pulsars with companion masses (greater than 0.005 $M_\odot$) and orbital periods (less than 1 day) are plotted. The companions that may be either low-mass (LM) stars or white dwarfs (WDs) are represented as hollow circles with a dot inside. Owing to their very low mass function, the companions of J1653-0158 and J1518+0204C are marked as low-mass stars. The data are taken from the Australia Telescope National Facility Pulsar Catalogue[34]. The mass ranges (grey horizontal lines) are calculated from the mass function, under the assumptions of a neutron star mass of 1.4 $M_\odot$ and an inclination angle of between 90 and 25.8 degrees (90% probability when assuming a random distribution). M71E with $P_b$ = 53.3 minutes and companion mass ($M_c$ approximately 0.05–0.10 $M_\odot$) is plotted as the red star with a black bar, with the median $M_c$ = 0.07 $M_\odot$. For comparison, a horizontal grey line with a star at the lower left corner represents the mass range of M71E when the orbital inclination angle is between 90 and 25.8 degrees. Blue curves represent the typical evolution tracks of the binary models[9], with an initial metallicity of the donor star of $Z$ = 0.02 (the solid line) or $Z$ = 0.002 (the dashed line). The red dashed–dotted curve represents the typical evolution of a low-mass X-ray binary with an evolved main-sequence donor, which will evolve into an ultra-compact X-ray binary eventually. The initial neutron star mass is assumed to be 1.40 $M_\odot$ in these models and other initial binary parameters are listed in the legend.

transfer. The orbital periods of such neutron star–He/CO white dwarf binaries can reach 53.3 minutes, at which point the He/CO white dwarf mass is supposed to be around 0.01 $M_\odot$ (ref. 29). This is in line with the companion mass constraint obtained from the mass function, as shown in Fig. 2. Furthermore, the orbital period of such a neutron star–He/CO white dwarf binary is expected to be around 6.80–9.64 minutes, which is significantly smaller than the orbital period of M71E, as the He/CO white dwarf mass decreases to 0.05–0.10 $M_\odot$. However, in this scenario, we do not consider the effect of evaporation by the pulsar emission, which may widen the binary separation and be helpful to explain the properties of M71E.

On the other hand, there are suggestions that at least some black widows could be the descendants of redbacks[8]. For these evolutionary tracks, the binary orbit initially shrinks because of magnetic braking and gravitational wave radiation, reaching a minimum in the orbital period before widening as a result of expelled, evaporated material by the strong pulsar wind. The system eventually reaches the black widow phase with a much smaller companion mass and moderate orbital period.

The small orbital period and moderate companion mass of M71E make it a bridging object between redbacks and black widows in the evolutionary track. To explain its parameters, we present two typical evolution tracks based on the binary evolution models[9]. The measured metallicity of M71 is [Fe/H] about −0.7 (that is, $Z$ = 0.002 (refs. 16,17)). To perform the calculations, we assume that the initial metallicity of the donor star is $Z$ = 0.002 (dashed line in Fig. 2, in case M71E is a member of M71) or $Z$ = 0.02 for solar metallicity (solid line in Fig. 2, in case M71E is not associated with M71, but is in the Galactic plane) (Methods). The low-metallicity model with an initial companion mass $M_{c,init}$ = 0.4 $M_\odot$ and orbital period $P_{b,init}$ = 0.70 days can roughly reproduce the position of M71E in the $M_c$–$P_b$ plot, even though the shortest orbital period from models is around 60 minutes, which is still longer than that of M71. The difficulty in explaining the observed orbital period may indicate that the composition of the companion is different from that in the model. Similar to some He-rich accreting white dwarfs known as cataclysmic variables[30], the companion star of M71E may be He rich, too. It is produced from the evolution of low-mass X-ray binaries with evolved main-sequence donors[31]. To illustrate this possibility, one evolutionary track of a low-mass X-ray binary with an evolved main-sequence donor was included in Fig. 2. This model is based on the binary evolution model with standard magnetic braking[22]. This model can reach the orbital period and mass of M71E. For more details of this model,

see the Methods section. In this model, the effect of evaporation by the pulsar emission, which should play an important role in the formation of spider pulsars and may widen the binary orbit, was not included. This model may also be helpful to explain these black widow pulsars with He-dominated donors, for example, the black widow pulsar PSR J1311-3430[32], a system with a 0.01 $M_\odot$ He-dominated companion star and an orbital period of 94 minutes. However, it is still unclear when the neutron star will appear as a millisecond pulsar and start to evaporate the companion.

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

# Methods

## Observation and data reduction

FAST surveys searching for new pulsars (see, for example, refs. 12,13,35) started in 2017, and about 800 new pulsars have been discovered so far[36]. M71E was first discovered by the FAST GPPS survey as PSR J1953+1844g in a binary system and identified as a possible member of M71 (ref. 12). It was later independently detected through the reprocessing of archival FAST globular cluster data obtained on 12 December 2019. With the same data, its orbital period was initially determined to be 53.3 minutes. We suggest that its formal name be PSR J1953+1846 (ref. 12) or M71E as it is fifthly identified and observed simultaneously with other pulsars in the M71. Follow-up observations have been arranged since September 2021.

We used the observations from MJD 59293 to 59781, in a total of 21 epochs. In each epoch, the observation lasted about 2 hr, covering more than twice the period of the orbit. All the observations were conducted with the FAST 19-beam L-band receiver, covering a frequency range of 1,050–1,450 MHz. The data were recorded in PSRFITS search mode format[37]. The number of frequency channels for the observations were either 4,096 (for a sampling time of 49.152 μs) or 2,048 (for a sampling time of 98.304 μs). More details of the observations can be found in either the FAST Globular Cluster Pulsar survey[13] or the GPPS survey[12].

Postprocessing of the data and timing analysis were carried out with the pulsar exploration and search toolkit (PRESTO)[38], TEMPO (https://tempo.sourceforge.net/) and TEMPO2[39] software packages. The routine 'prepfold' in PRESTO was used for folding the search-mode data and creating the archives of integrated profile, each with integration time of 5 minutes. The number of phase bins of the pulse profile is 128. The routine 'get_TOAs.py' in PRESTO was then used to obtain the times of arrival of these pulse profiles. For each observation, typically 32–64 times of arrival were obtained to track the phase of the orbit. During the timing analysis, the JUMPs (that is, the time intervals added between two sets of arrival times from two observations) were added between each pair of observations so that the orbital period, epoch of periastron passage and projected semimajor axis can be initially fitted. The ELL1 binary model was used, aiming to fit a possibly very small eccentricity of the orbit. As many as possible of these JUMPs were then removed to derive the spin frequency, spin frequency derivative and the coordinates of this pulsar. The code determining the rotation count of pulsars (DRACULA)[40] was used for removing these JUMPs. The search data were folded again with the timing solution to obtain the times of arrival. The data-processing steps were iterated until the timing solution agreed with the previous one. Extended Data Fig. 1 shows the timing residuals achieved in the end, after subtracting the best-estimated timing solution shown in Table 1. No eclipsing events were observed. Thus, the timing residuals around the orbit phases were flat (see middle panel of Extended Data Fig. 1). Two-hour observation data (more than two orbits) were also folded to show that the flux of the pulsar signal did not change along the orbital period (see lower panel of Extended Data Fig. 1).

Extended Data Fig. 2 shows the polarization pulse profile of the pulsar on 4 December 2021. The integration time of the observation is 3,600 s and the observational frequency band is 1,050–1,450 MHz. The polarization calibration was conducted using the scan adjacent to the pulsar observation on the periodic signal of noise diode with a temperature of 10 K. The aperture efficiency 0.63 of FAST was adopted in the calibration process[41]. Data were processed (including the folding, dedispersing and calibrating processes) with the software laconic program units for pulsar data analysis (LAPUDA; https://github.com/lujig/lapuda) and the ephemeris parameters in Table 1. The rotation measure (RM) was obtained by searching in the range from −1,000 to 1,000 rad m$^{-2}$, and was determined to be −475 ± 2 rad m$^{-2}$. The peak and mean flux of the total intensity of M71E were estimated to be 0.632 ± 0.002 mJy and 0.092 ± 0.002 mJy, respectively.

## Multiband data

The optical or infrared counterpart was not found on the SDSS and 2MASS images, whereas the Chandra data show a possible X-ray counterpart. In the Chandra data with observation ID 5434 (https://cda.cfa.harvard.edu/chaser/mainEntry), eight photons can be found in a 0.5″-radius circle near the position (J2000, RA J2000 19:53:45.2865; dec. +18:44:54.5075), which is 7.34 milliarcseconds away from the timing position. With the ephemeris parameters in Table 1, the corresponding pulsar rotating phases of the arrival times for the eight photons were −0.184, −0.143, 0.478, 0.050, −0.096, 0.123, 0.036 and −0.093. Apart from the third photon, the other seven photons clustered in a phase interval from −0.184 to 0.123. Although the possibility of such distribution for random photons is only about 3%, it is believable that these photons came from M71E. The energies of the eight photons are 0.838, 0.759, 1.045, 0.765, 1.231, 1.327, 0.966 and 1.003 keV. The X-ray flux can be calculated as $3 ± 1 × 10^{-16}$ erg cm$^{-2}$ s$^{-1}$. Assuming that all the photons came from M71E and that its distance is the same as that of M71 (4 kpc), the X-ray luminosity of this source can be estimated as $6 ± 2 × 10^{29}$ erg s$^{-1}$, that is, 12 ± 4 nCrab, which is broadly consistent with the luminosities observed in other non-accreted millisecond pulsars[42], implying that the X-ray may be rotation-powered and that there is an absence of, or very weak, accretion flow. The X-ray luminosity of the pulsar would induce a donor temperature of about 3,000 K for an equilibrium state, which is approximately the temperature of the brown dwarf in the LMX or LMZ model[23]. Therefore, even if the donor is a white dwarf, the entire star would be heated by the pulsar to that equilibrium temperature. With such a high temperature, the mass–radius relation of the white dwarf would largely deviate from the zero-temperature case, and may be similar to that of the brown dwarf. Consequently, it may be a little hard to distinguish the donor with information on mass and radius only; optical spectra and imaging may help in the future.

To narrow down which evolutionary channel M71E originated from, monitoring observations in multibands are necessary. Given the theoretical properties (luminosity, $3.25 × 10^{-3} L_\odot$; effective temperature, roughly 4,500 K) from our model and the distance of M71E (4 kpc), the apparent magnitude of the companion star was estimated to be $m_v = 25.1$ mag. This is much lower than the limits of the archival optical data from SDSS (about 22 magnitude at all five bands[43]). It should be possible to obtain its optical spectra with the Hubble Space Telescope with its Space Telescope Imaging Spectrograph of G430M/G750M gratings (or even the more powerful JWST), covering a wavelength range of 3,050 to 10,100 Å, with a resolving power of about 17,000. With the spectra, we may infer its element abundances, which will be helpful to know the possible type of the donor, which may be from an evolved main-sequence star. Previous detections of the companions of eclipsing black widows were done with the Hubble Space Telescope for M71A[44] (in the same globular cluster M71) and M5C[45]. As the evolution routine of a white dwarf companion is also possible, the optical detection of the companion may also conclude the type of the companion. If the companion mass can also be derived, the pulsar mass and orbital inclination angle will be obtained correspondingly. Continuing timing with radio telescopes could yield the orbital derivative, parallax and proper motion of this binary. In addition, in such a compact orbit, the interaction between the pulsar and the companion should be active. This will be seen in detail by observing in wider radio bands, and with simultaneous observations in radio and X-ray.

## Details of the evolutionary models applied in this paper

The theoretical models[9] are indicated by the blue curves in Fig. 2. In Extended Data Fig. 3, we show the evolution of a binary system with an initial neutron star mass $M_{p,init} = 1.40 M_\odot$, an initial companion mass $M_{c,init} = 0.40 M_\odot$, an initial orbital period $P_{b,init} = 0.70$ days and an evaporation efficiency $f = 0.02$. The initial metallicity of the companion star is $Z = 0.002$. At the early phase of the evolution, the magnetic braking

dominates the angular momentum loss, leading to the decrease of binary separation. At $t = 9.35$ Gyr, the companion star fills its Roche lobe and starts mass transfer. When the secondary mass decreases to around 0.30 $M_\odot$, we find that the donor star becomes fully convective and the magnetic braking ceases to operate, leading to the detachment of the binary system. At this time, we assume that the neutron star becomes a radio pulsar and the emission from the pulsar starts to evaporate the companion star. From this point, the evolution of binary separation is mainly influenced by the competition between the shrinking of the orbit because of gravitational wave radiation and the widening of the orbit because of expelled, evaporated material. At the beginning of the second phase, the gravitational wave radiation dominates the angular momentum loss, reaching a minimum in the orbital period (about 60 minutes). Afterwards, the binary separation starts to increase because of expelled, evaporated material.

The binary evolutionary model indicated by the red curve in Fig. 2 is based on the models with standard magnetic braking prescription[22]. The initial neutron star and donor masses in this example are 1.40 $M_\odot$ and 1.00 $M_\odot$. The initial orbital period is 2.25 days. The initial metallicity of the donor star is $Z = 0.002$. In Extended Data Fig. 4, we present the evolution of the donor star in the Hertzsprung–Russell diagram, the evolution of mass transfer rate as a function of orbital period and the evolution of surface He abundance. Before the onset of mass transfer, the orbital period decreases because of magnetic braking. When the orbital period decreases to about 0.62 days, the donor star located around the end of the main sequence starts to transfer material onto the neutron star. As the donor star loses its mass, its core is exposed and surface He abundance increases. When the orbital period is around 53 minutes, the donor mass is around 0.07 $M_\odot$ and the surface He abundance is around 0.76. As the H-rich envelope is fully stripped, the binary system evolves into an ultra-compact X-ray binary and the orbital period increases.

## Data availability

According to the data policy, the FAST observational data related to this work would be public roughly 1 year after the observation. Public data, including partial M71E data related to this paper, are available and can be obtained by sending a request to the FAST data centre. Details of the FAST data related to this study and the FAST public data list can be found at http://fast.bao.ac.cn.

## Code availability

The code used to search for pulsars in M71 is PRESTO[38,46] and the Re-analysing Pipeline for Parkes Pulsar Survey (RPPPS)[47], which uses PRESTO in parallel. Timing was done with routines in PRESTO, TEMPO and TEMPO2. The custom code to process FAST pulsar search data, especially for polarizations, is named LAPUDA. These codes can be found at: PRESTO (https://github.com/scottransom/presto); RPPPS (https://github.com/qianlivan/RPPPS); TEMPO (https://tempo.sourceforge.net/); TEMPO2 (https://www.atnf.csiro.au/research/pulsar/tempo2/); DRACULA (https://github.com/pfreire163/Dracula); and LAPUDA (https://github.com/lujig/lapuda).

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

**Acknowledgements** This work is supported by the Chinese Academy of Sciences Project for Young Scientists in Basic Research, grant no. YSBR-063; National Key Research and Development Program of China nos. 2018YFE0202900, 2021YFA1600403 and 2022YFC2205202; Square Kilometre Array (SKA) Program of China no. 2020SKA0120200; and National Science Foundation, China, nos. 12225303, 12288102, 11703047, 11773041, 11833058, 11873058, 11988101, 12003047, 12041303, 12073071, 12133004, 12173052, 12173053, 12273072, U1531246, U2031119 and U1931128. P.J. is supported by the Chinese Academy of Sciences Project for scientific instruments, no. YJKYYQ20200021, and Guizhou high-level innovative talents, no. GCC[2022]003-1. Z.P., L.Q. and H.-L.C. are or were supported by Chinese Academy of Sciences 'Light of West China' Program. Z.P. and L.Q. are or were supported by the Youth Innovation Promotion Association of the Chinese Academy of Sciences (ID nos. 2023064 and 2018075). M.H.L. is supported by the Guizhou Provincial Basic Research Program (Natural Science, KZ[2023] 039). This work has been supported by the New Cornerstone Science Foundation through the Xplorer Prize. This work made use of data from the FAST. FAST is a Chinese national mega-science facility, built and operated by the National Astronomical Observatories, Chinese Academy of Sciences (NAOC). We thank all the people from the FAST group for their support and assistance during the observations. We thank T. Tauris for useful discussions on the formation scenario of the system. H.-L.C. thanks L. Wang and J. Li for helpful discussions on the possibility of optical observation. This research has made use of data obtained from the Chandra Data Archive and the Chandra Source Catalog, and software provided by the Chandra X-ray Center (CXC) in the application packages CIAO and Sherpa.

**Author contributions** Z.P., J.G.L., H.-L.C., P.J., J.L.H. and B.Z. led the study, including proposing and arranging the FAST observations, and finding a possible way to explore the origin of the pulsar. Z.P., J.G.L., K.L., D.J.Z. and Z.L.Y. processed the data, searched for the pulsar signal and obtained the timing solution. H.-L.C., Z.W.H., J.G.L., K.L. provided the initial model for the binary pulsar and used the models for this work. Z.W.H., K.L., R.X.X., B.Z., L.Q., J.T.L., Z.Y., Z.L.Y., D.J.Z., P.F.W. and C.W. participated in the investigations, including observation, searching for the signal of M71E, timing the pulsar and studies based on pulsar timing results. Z.P., J.G.L., H.-L.C., Z.W.H., K.L., R.X.X., B.Z., P.J. and J.L.H. contributed towards methods for data reduction, analysis and modelling. The project was administrated by P.J., Z.P., J.G.L., H.-L.C., J.L.H., M.Z. and M.H.L. All the authors contributed to writing the initial manuscript and to modifications.

**Competing interets** The authors declare no competing interests.

**Additional information**
**Correspondence and requests for materials** should be addressed to P. Jiang, J. L. Han or B. Zhang.

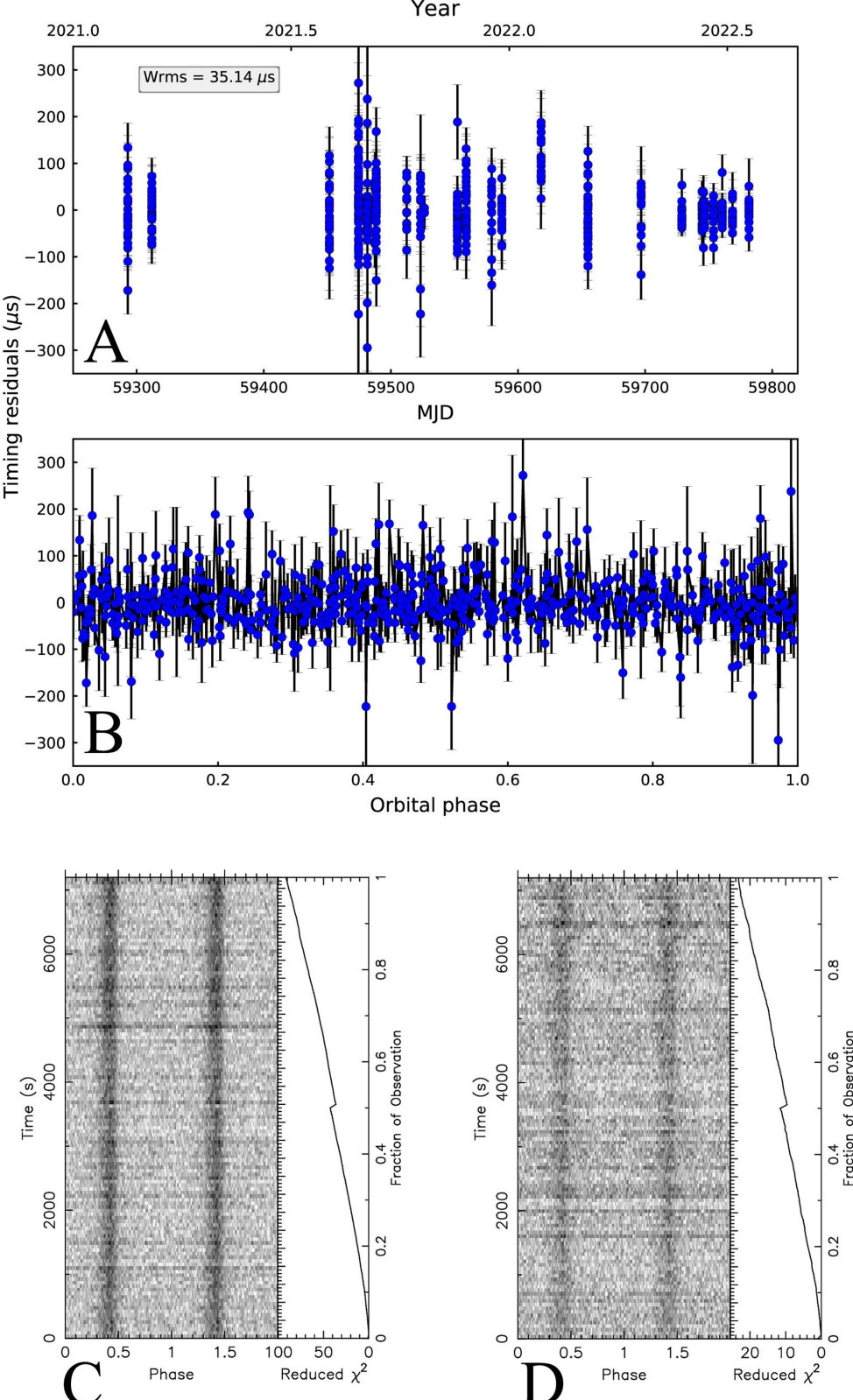

**Extended Data Fig. 1 | Timing residuals and lack of eclipsing of M71E.** Panel a: Timing residuals of M71E, obtained by subtracting the best-estimated timing model shown in Table 1 from the time-of-arrivals. Panel b: timing residuals as a function of orbital phase, showing that there is no eclipsing events. Panel c and d: as an example, from the observation in December 4[th], 2021, the data were folded with timing solution. The observation covered more than 2 orbits, while no eclipsing was seen either for the whole FAST band (1.05–1.45 GHz, panel c), or the lowest 1.05–1.10 GHz subband (panel d). All the error bar are for ± 1 sigma.

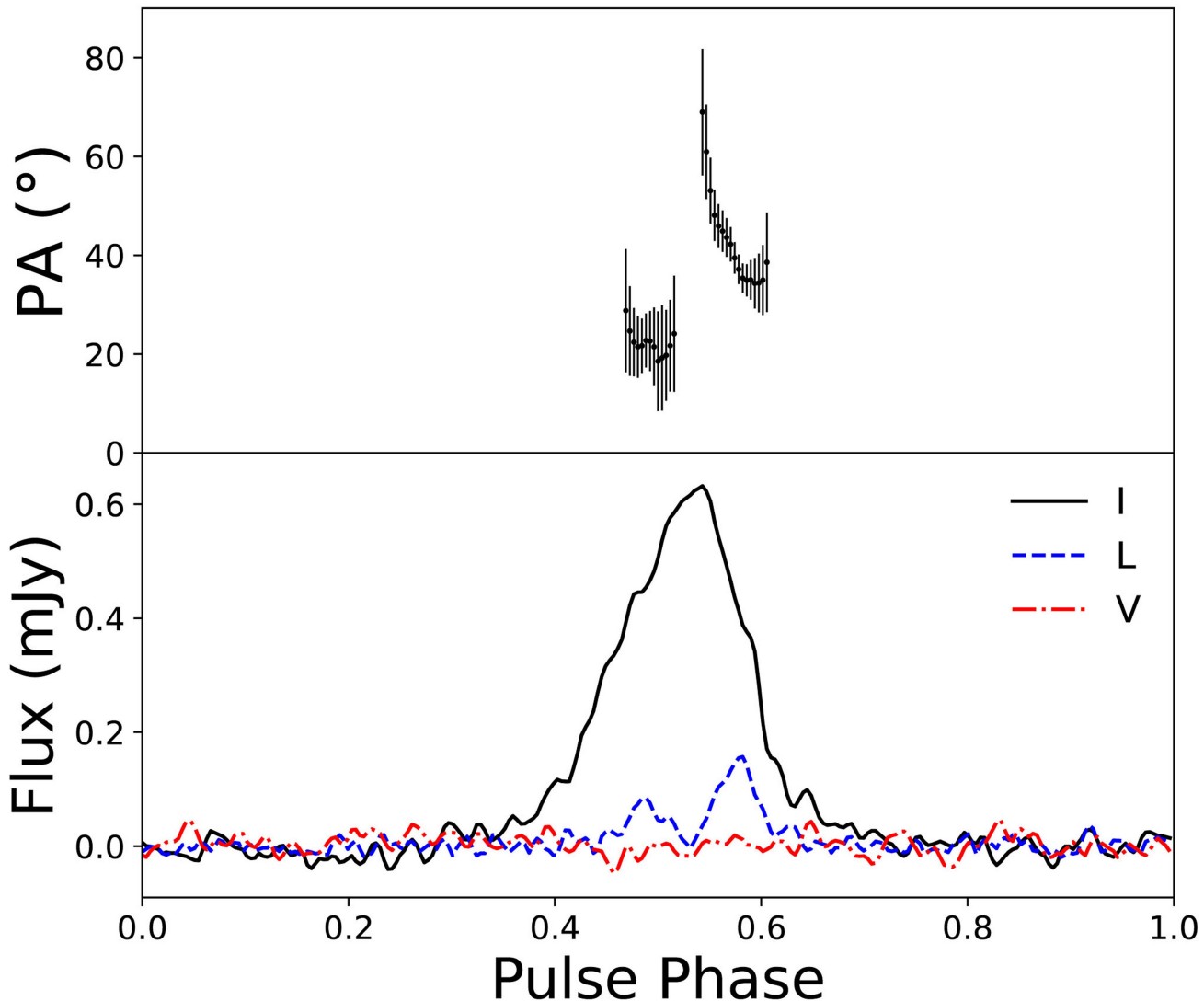

**Extended Data Fig. 2 | Polarization pulse profile of M71E.** The intensity of the flux, linear polarization and circular polarization are presented as black solid, blue dashed and red dash-dotted curves, with polarization position angles with ±1 sigma error bar above. The peak and mean flux of the total intensity of M71E was estimated to be $0.632 \pm 0.002$ and $0.092 \pm 0.002$ mJy, respectively. The RM used to obtain this pulse profile is $-475 \pm 2$ rad m$^{-2}$.

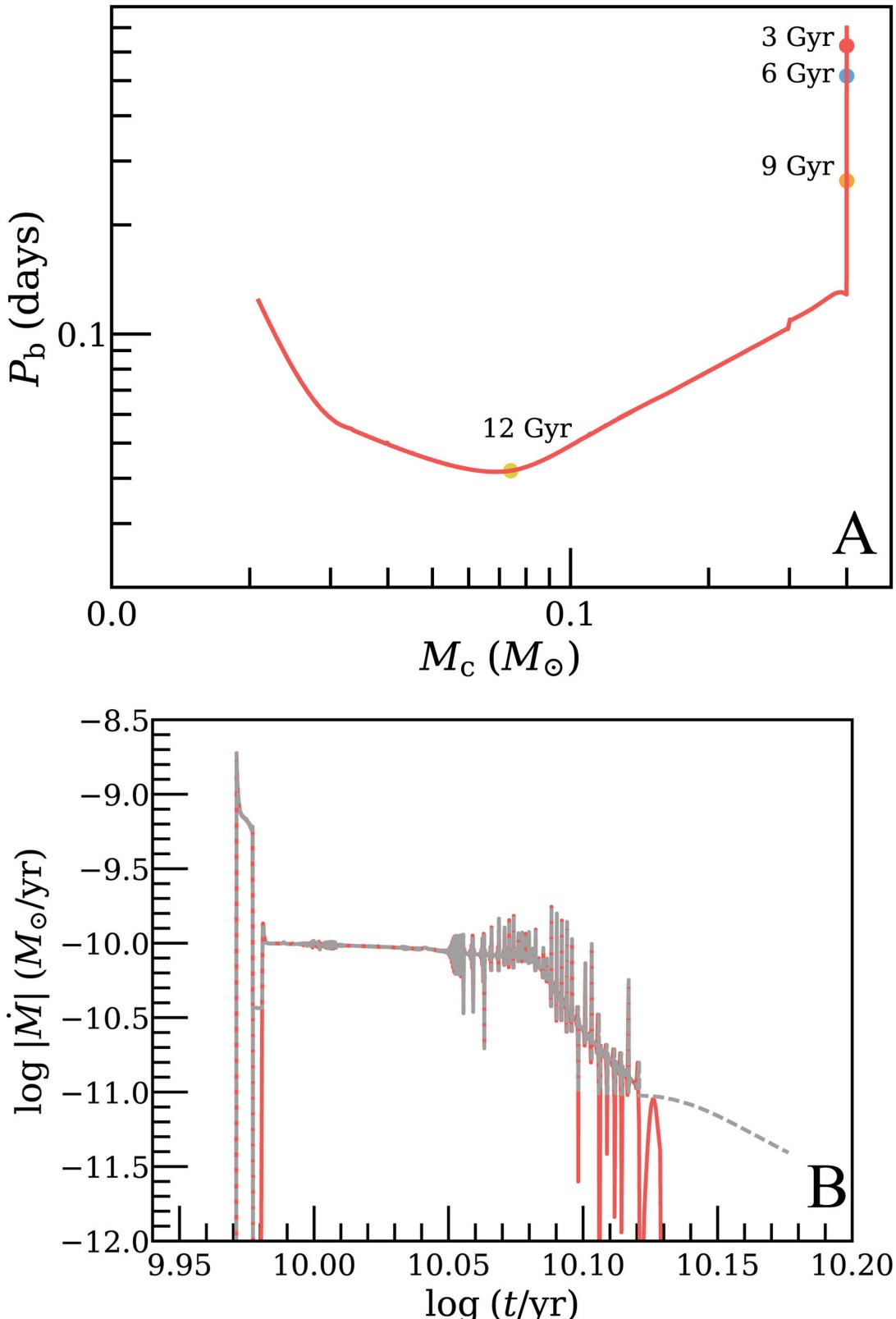

**Extended Data Fig. 3 | Evolution of a binary system with an initial neutron star mass $M_{p,init}$ = 1.40 $M_{\odot}$, an initial companion mass $M_{c,init}$ = 0.40 $M_{\odot}$ and an initial orbital period $P_{b,init}$ = 0.70 days.** Panel a: Evolution of donor mass and orbital period. The ages of the system at different epochs are indicated in the plot. Panel b: Evolution of mass transfer rate (red line) and total mass-loss rate (grey dashed line) as a function of time.

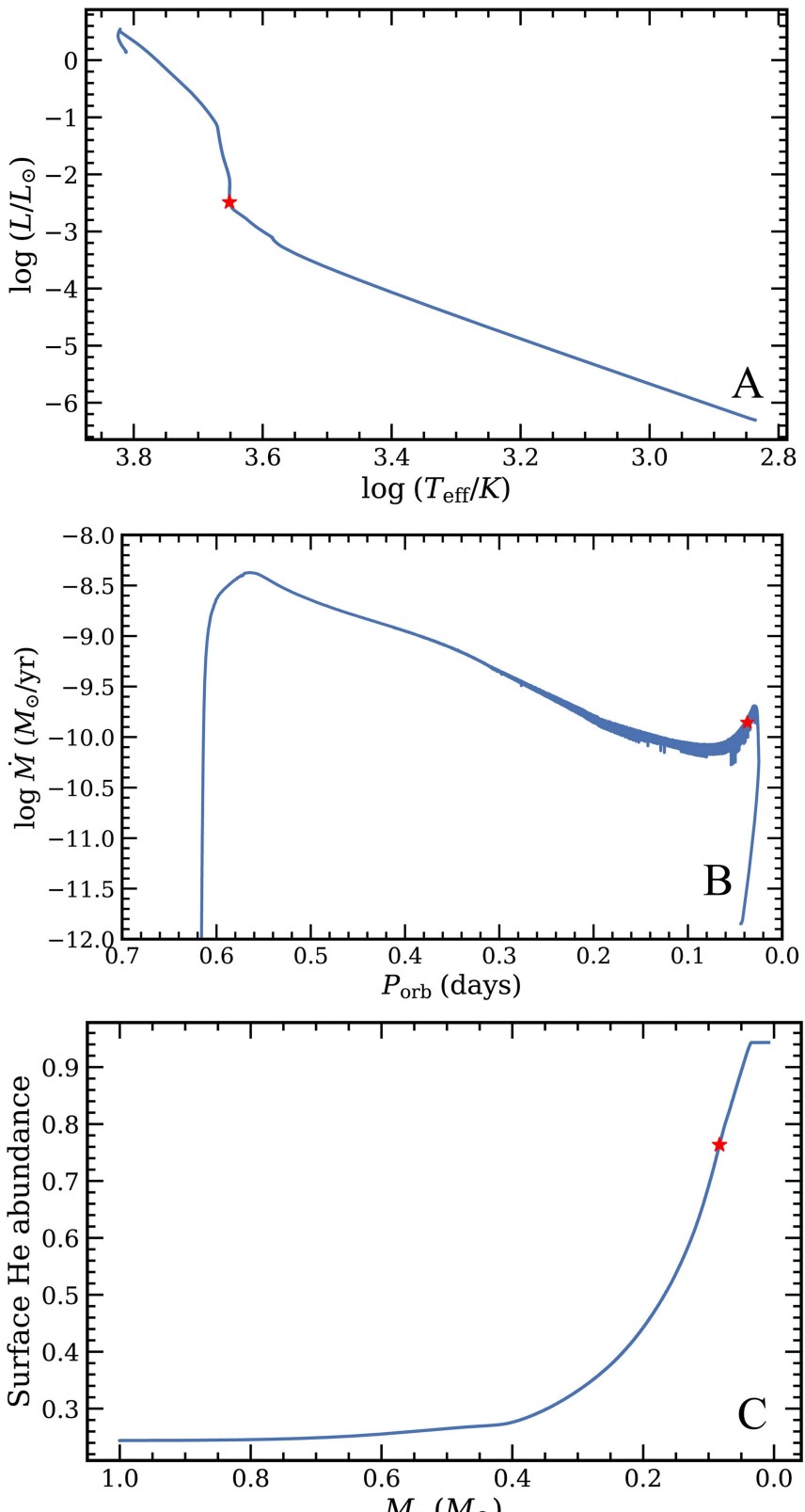

**Extended Data Fig. 4 | Details of the donor evolution.** Panel a: evolution of the donor star in the HR diagram; Panel b: evolution of mass transfer rate as a function of orbital period; Panel c: evolution of surface He abundance of the donor star as a function of donor mass. The initial binary parameters for this binary system are $M_{p,init} = 1.40\,M_\odot$, $M_{c,init} = 1.00\,M_\odot$ and $P_{b,init} = 2.25$ days. The red stars indicate the models with an orbital period same as the orbital period of M71E, i.e. 53 min.