## [Peer Review File · Nature]

Manuscript Title: A Binary Pulsar in a 53-minute Orbit

Reviewer Comments & Author Rebuttals

Reviewer Reports on the Initial Version:

Referees' comments:

Referee #1 (Remarks to the Author):

This paper describes the detection of a pulsar in a 53 minute orbit with a low mass companion. This discovery is significant in understanding the evolution of the most compact spider binaries, and carries significant value because it was made through the direct detection of the pulsar. In my opinion, the discovery carries enough weight that it could warrant publication in Nature, especially given the immense value of a pulsar detection. However, I believe the manuscript needs significant work, and I outline some items below which I believe should be addressed. I start with specific comments on the text, and then general comments about what I think is missing:

Bold faced paragraph:

“Observed for tens, these systems are key for studying the evolutionary link between accreting X-ray pulsars and isolated millisecond pulsars, effects of pulsar irradiation, and the birth of massive neutron stars”

>Typo in the first clause?

>Also, I would urge the authors to cite Romani et al. 2022, a very high-profile result which made a robust measurement of a 2.35 solar mass neutron star in a black widow binary, that has really pushed on the equation of state constraints for nuclear matter (that is one of the main reasons these objects are so physically exciting). I was quite surprised to not see a single paper by Roger Romani or Jay Strader among the references here, given the role they have played in the observational study of these systems in recent years.

>I think it is worth emphasizing in the bold faced paragraph that this system resides in a globular cluster. That may be a significant clue in the formation of these objects (as globular clusters allow for dynamical formation scenarios which are much less likely to occur in the field).

“PSR J1953+1846E (M71E) was firstly discovered in 2021 using the Five-hundred-meter Aperture Spherical radio Telescope⁹, ¹⁰ by the FAST Galactic Plane Pulsar Snapshot survey¹¹. It was later identified to be in a compact binary system from the archival globular cluster data¹² .”

>I would like to understand this (as someone who doesn't work a lot on radio data). I was under the impression that detecting a pulsar in such a compact orbit is quite challenging because the objects are highly accelerated, and require coherent acceleration searches to detect. This statement suggests that the source was first discovered, and later identified as residing in a binary. My question is, how could one discover the source without automatically knowing that it is in a binary, given that the pulses should exhibit significant time delays on short timescales due to the compact orbit? (e.g., wouldn't one need to use an acceleration search over a range of orbital periods and potential projected semi-major axes to pick up the signal, at which point, wouldn't one already know the exact orbital properties of the system?). Or was the initial discovery a detection in a continuum image? I apologize if I'm being naïve here, as someone who has not worked extensively with radio pulsation data.

“Its timing position (J2000 19:53:37.95 +18:44:54.3) is approximately 2.5 arcminutes away from the center of M71 (J2000 19:53:46.49 +18:46:45.1, the 2010 edition¹⁵). Therefore, it is debatable whether this pulsar is a member of this globular cluster”

>It might be worth noting that if it is a member of the cluster, it is on the outskirts (which I believe would imply it is an old system).

"The tight orbit limits the possibility of the companion star being a dwarf star."

>You could just say that the tight orbit eliminates the possibility of a main sequence star, with only evolved donors, white dwarfs, or brown dwarfs being dense enough to reach such a period.

". A possible explanation is that the companion star is a He-rich companion, produced from the evolution of low mass X-ray binaries with evolved main sequence donors"

>Could be worth citing some of the examples of this channel among CVs (e.g. the 51 min system with an evolved donor that was published just recently). It seems to be this channel could be very viable for producing such systems.

Fig 1:

>I think the authors should make it clear that the WD mass limit is a "lower limit" in the legend. Also, is it clear that this system couldn't have hosted an ELM WD which overflowed its roche lobe and started undergoing stable mass transfer and then was driven to lower masses as a result? I believe the progenitor of such a system was recently published by Sam Swihart.

>In the lower panel, maybe it's worth pointing out that the radius of the companion as a function of mass follows straightforwardly from the density vs period relation? (e.g. $P_{\text{RLO}}(\text{days}) = 0.43/\sqrt{\rho(\text{g/cm}^3)}$). The mass ratios here are less than 100, so that relation should hold for all scenarios.

Fig 2: Would it be possible to include a track with a He-enriched evolved donor? Presumably, those have little trouble reaching these periods/masses.

Broad comments:

This paper is entirely based on a radio detection, which while compelling, is only a small aspect of understanding these systems. The paper does not discuss searches for an optical/IR counterpart, an x-ray/gamma-ray counterpart, etc. At the very least, there should be discussion of the feasibility of such observations (presumably the source is very well localized in the radio). For example, I took a quick cursory glance at the Chandra image of this field (with the source coordinates at the center of the image where those purple crosshairs are slightly below the middle of this screenshot), and it looks to me like there is an x-ray counterpart there, which is exactly what one would expect for such a source, since the intra-binary shock tends to emit x-ray flux in these systems. This is just one example of where I think the paper could use more work—there is not even mention of x-rays or Chandra in the manuscript, yet the data is sitting right there. I would expect a publication in Nature to have been thorough about checking archival data at all wavelengths, and not simply focusing exclusively on a radio detection.

I also think the authors should discuss a broader range of evolutionary scenarios. The proximity to the globular strongly suggests an association, and this has evolutionary implications such as formation via dynamical interactions. Additionally, it is worth spending a little bit more time discussing the evolved donor possibility, especially since this is a channel that is proving to be important in the formation of other types of ultracompact binaries.

Finally, it would be useful for the paper to discuss future prospects in further characterizing this system, to help narrow down on which evolutionary channel it originated from. Could further timing be used to constrain the presence of a wide companion in a triple (which might have resulted in Kozai-Lidov cycles?). Could phase-resolved spectroscopy in the optical reveal the composition of the donor (and would this be feasible with current optical/IR telescopes given the luminosity of the object and a nominal distance of 4 kpc to M71?). Could a deeper x-ray observation reveal something about the nature of the system which is not apparent in the radio? Exploring questions like these would help make the discovery feel more compelling to the broader community, whereas currently, this is a manuscript that feels to me as though it is targeting the

radio pulsar community (and the goal of a journal like Nature is to aim for broader appeal than that).

Referee #2 (Remarks to the Author):

Nature 2023-03-03727

Pan et al.

"A Binary Pulsar in a 53-minute Orbit"

This is a very interesting, and quite straight-forward, paper about the discovery of the shortest orbital period binary pulsar. The orbital period is 30% shorter than the previous record holder, a gamma-ray black widow system uncovered only a couple years ago.

That alone makes the pulsar quite interesting. But the fact that it also seems to be in a mass range between so-called black-widow and redback pulsars implies a possible evolutionary link between those similar but separate "spider" systems. That does potentially tell us something new about binary evolution.

The one possible caveat to that is the fact that this system *might* be either from, or currently a part of, globular cluster M71. And the stellar dynamics in globular clusters can do crazy things.

Nevertheless, this is a Nature-worthy result, and the paper describes it all quite nicely.

I have only a few relatively minor comments.

-- Nowhere in the paper do the authors discuss whether this pulsar exhibits radio eclipses, which leads me to believe that it does not. Given that "spider" pulsars often eclipse (especially the redbacks), the fact that this one likely does not is surprising. I think this should be explicitly discussed, if only for a couple sentences.

The lack of eclipses (assuming that is correct!) also strengthens the arguments in the paper that suggest that the system is viewed nearly face-on. It is worth mentioning for that point alone.

If space allowed (even just in the methods) it might be valuable to show either the timing residuals as a function of orbital period (which should be distributed uniformly without eclipses) or a pulse phase vs time greyscale (as shown on a prepfold plot, for instance) for a long observation which covered >1 full orbit. This would especially be useful at the lowest observed frequencies, since eclipses are most apparent there.

-- I would request that the authors include an estimated flux density of the pulsar. That is important for others who might try to observe the system.

-- Summary: "Observed for tens, ..." Is that a typo? What does it mean?

-- Fig 2 caption: "with the medium" should be "with the median", I think

-- Fig 2: This figure seems to be missing J1653-0158, the previous record holder (described in Nieder et al 2020), with PB = 0.052 days. (Note that I'm not as concerned with the other *likely* pulsar mentioned in the first paragraph, FTZ J1406+1222).

That's all I have. Very nice result!

Scott Ransom

Author Rebuttals to Initial Comments:

We thank both referees for very helpful suggestions and comments on the manuscript. These suggestions and comments have greatly helped us to improve the presentation of the paper. We have carefully revised the manuscript accordingly to address all these comments. All the revisions are marked in blue. The deleted text is marked with deletion marks. In the following, we provide point-to-point replies to each of the referees' comments:

Referees' comments:

Referee #1 (Remarks to the Author):

This paper describes the detection of a pulsar in a 53 minute orbit with a low mass companion. This discovery is significant in understanding the evolution of the most compact spider binaries, and carries significant value because it was made through the direct detection of the pulsar. In my opinion, the discovery carries enough weight that it could warrant publication in Nature, especially given the immense value of a pulsar detection. However, I believe the manuscript needs significant work, and I outline some items below which I believe should be addressed. I start with specific comments on the text, and then general comments about what I think is missing:

--We thank the referee for acknowledging the significance of the discovery and for providing many suggestions to improve the paper.

Bold faced paragraph:

“Observed for tens, these systems are key for studying the evolutionary link between accreting X-ray pulsars and isolated millisecond pulsars, effects of pulsar irradiation, and the birth of massive neutron stars”

>Typo in the first clause?

--Thanks for pointing out the typo. It has been corrected.

>Also, I would urge the authors to cite Romani et al. 2022, a very high-profile result which made a robust measurement of a 2.35 solar mass neutron star in a black widow binary, that has really pushed on the equation of state constraints for nuclear matter (that is one of the main reasons these objects are so physically exciting). I was quite surprised to not see a single paper by Roger Romani or Jay Strader among the references here, given the role they have played in the observational study of these systems in recent years.

--Thanks for drawing us attention to these important and relevant papers. We have read them and added the citations to these papers in the first paragraph and also in the 8th paragraph of the main text.

>I think it is worth emphasizing in the bold faced paragraph that this system resides in a globular cluster. That may be a significant clue in the formation of these objects (as globular clusters allow for dynamical formation scenarios which are much less likely to occur in the field).

--Thanks. We have revised the boldfaced paragraph to state that this binary is only several arcminutes away from the center of M71.

“PSR J1953+1846E (M71E) was firstly discovered in 2021 using the Five-hundred-meter Aperture Spherical radio Telescope^{9, 10} by the FAST Galactic Plane Pulsar Snapshot survey¹¹. It was later identified to be in a compact binary system from the archival globular cluster data¹².”

>I would like to understand this (as someone who doesn’t work a lot on radio data). I was under the impression that detecting a pulsar in such a compact orbit is quite challenging because the objects are highly accelerated, and require coherent acceleration searches to detect. This statement suggests that the source was first discovered, and later identified as residing in a binary. My question is, how could one discover the source without automatically knowing that it is in a binary, given that the pulses should exhibit significant time delays on short timescales due to the compact orbit? (e.g., wouldn’t one need to use an acceleration search over a range of orbital periods and potential projected semi-major axes to pick up the signal, at which point, wouldn’t one already know the exact orbital properties of the system?). Or was the initial discovery a detection in a continuum image? I apologize if I’m being naïve here, as someone who has not worked extensively with radio pulsation data.

—A search for binaries may extend the parameter space from 2-dimension (period and dispersion) to 7-dimension (5 parameter to describe the orbit, or at least 3 parameters to describe a circular orbit). The acceleration search using period derivative to describe the orbital movement saves a lot of time. Such searches can identify acceleration rather than constraining orbital parameters. Accordingly, the limitation of acceleration search is that the observation time should be much shorter than the orbital period, normally 10% or less of the orbital period.

For finding this pulsar, both GPPS and FAST globular cluster pulsar survey used PRESTO in which the coherent acceleration search (PRESTO routine *accelsearch*) was applied. The GPPS project uses 5 minutes observation time for discovery and follow-up observations. Such a short observation time makes the detection to this binary possible, while difficult to obtain the orbital parameters.

FAST globular cluster (GC) pulsar survey has archival data of M71 for 5 hours. The observations were to the center of the GCs. As M71E is ~ 2.5 arcminutes away from the center and the beam size of FAST at L-band is around 3 arcminutes, the pulsar signal in FAST GC pulsar survey data is very faint. With efforts in filtering faint pulsar candidates, the pulsar was luckily detected in the 5-hour data and thus its orbital parameters were determined.

We rewrote the sentences to make it clear.

“Its timing position (J2000 19:53:37.95 +18:44:54.3) is approximately 2.5 arcminutes away from the center of M71 (J2000 19:53:46.49 +18:46:45.1, the 2010 edition¹⁵). Therefore, it is debatable whether this pulsar is a member of this globular cluster”

>It might be worth noting that if it is a member of the cluster, it is on the outskirts (which I believe would imply it is an old system).

—Thanks, we have added a description that if at the similar distance from M71, M71E is in the outskirts of the globular cluster.

“The tight orbit limits the possibility of the companion star being a dwarf star.”

>You could just say that the tight orbit eliminates the possibility of a main sequence star, with only evolved donors, white dwarfs, or brown dwarfs being dense enough to reach such a period.

--Thanks for pointing this out. Corrected.

“ A possible explanation is that the companion star is a He-rich companion, produced from the evolution of low mass X-ray binaries with evolved main sequence donors”

>Could be worth citing some of the examples of this channel among CVs (e.g. the 51 min system with an evolved donor that was published just recently). It seems to be this channel could be very viable for producing such systems.

--We have included the reference for the He-rich CV (ZTF J1813+4251), which also has a He-rich donor star.

Fig 1:

>I think the authors should make it clear that the WD mass limit is a “lower limit” in the legend. Also, is it clear that this system couldn’t have hosted an ELM WD which overflowed its roche lobe and started undergoing stable mass transfer and then was driven to lower masses as a result? I believe the progenitor of such a system was recently published by Sam Swihart.

--We have corrected the legend. In addition, we can exclude the possibility that this system is evolved from a NS + ELM WD binary. In Chen et al. (2022), we have modelled the evolution of NS + He WD binaries with stable mass transfer. From their results, we expect that the orbital periods of these NS-He WD binaries are around 6.80–9.64 min, significantly smaller than the orbital period of M71E, as the WD masses decrease to 0.05 – 0.10 Msun. We have included the discussion on this point in the 7th paragraph.

>In the lower panel, maybe it’s worth pointing out that the radius of the companion as a function of mass follows straightforwardly from the density vs period relation? (e.g. $P_{RLO}(\text{days})=0.43/\sqrt{\rho(\text{g/cm}^3)}$). The mass ratios here are less than 100, so that relation should hold for all scenarios.

--Thanks for reminding us to explain the limit in the lower panel of Figure 1. In the manuscript, the black curve in the lower panel is calculated with the relation (Eggleton 1983)

$$R_{RL} = a \frac{0.49q^{\frac{2}{3}}}{0.6q^{\frac{2}{3}} + \ln\left(1 + q^{\frac{1}{3}}\right)},$$

where R_{RL} is the Roche lobe radius, a is the distance between the pulsar and the donor, and q is the mass ratio. When q is much less than 1, e.g., in M71E’s situation, this equation is almost as same as that given by the referee. We also added a sentence in the text to describe this figure in the main text.

Fig 2: Would it be possible to include a track with a He-enriched evolved donor? Presumably, those have little trouble reaching these periods/masses.

--In Fig. 2, we have added one evolutionary track for a low mass X-ray binary with an evolved main sequence donor. This track can reach the orbital period and mass of M71E. In Extended Data Fig. 4, we have more detailed description for this model.

Broad comments:

This paper is entirely based on a radio detection, which while compelling, is only a small aspect of understanding these systems. The paper does not discuss searches for an optical/IR counterpart, an x-ray/gamma-ray counterpart, etc. At the very least, there should be discussion of the feasibility of such observations (presumably the source is very well localized in the radio). For example, I took a quick cursory glance at the Chandra image of this field (with the source coordinates at the center of the image where those purple crosshairs are slightly below the middle of this screenshot), and it looks to me like there is an x-ray counterpart there, which is exactly what one would expect for such a source, since the intra-binary shock tends to emit x-ray flux in these systems. This is just one example of where I think the paper could use more work—there is not even mention of x-rays or Chandra in the manuscript, yet the data is sitting right there. I would expect a publication in Nature to have been thorough about checking archival data at all wavelengths, and not simply focusing exclusively on a radio detection.

--Prompted by the referee's important remark, we have checked the archival data from Fermi, Chandra, SDSS and 2MASS data for M71E, and added a description of this binary in the multiband archival data in the main text and Methods. We confirm the referee's work that there is an obvious x-ray counterpart in the Chandra data. However, we did not find any counterpart in the Fermi, SDSS (optical), and 2MASS (IR) archival data.

I also think the authors should discuss a broader range of evolutionary scenarios. The proximity to the globular strongly suggests an association, and this has evolutionary implications such as formation via dynamical interactions. Additionally, it is worth spending a little bit more time discussing the evolved donor possibility, especially since this is a channel that is proving to be important in the formation of other types of ultracompact binaries.

--In the 7th paragraph, we have now included some discussion on the dynamical formation scenario. Based on the previous studies, e.g. King et al. (2003, 2005), we think that producing from such a scenario is not impossible. However, due to the lack of self-consistent and detailed calculation on this formation scenario, it is hard to confirm such a channel.

In the 8th paragraph, we have added more discussion on the evolved donor possibility. We have a more detailed description on this scenario. We also think that this scenario may be helpful to explain these spider pulsars with He-rich companions, such as PSR J1311-3430. This spider pulsar has a He companion star and an orbital period of 91 min. However, we have also addressed that we did not include the effect of evaporation by the pulsar emission, since it is still

unclear when the NS will appear as a pulsar and start to evaporate the donor star. This aspect deserves a further study.

Finally, it would be useful for the paper to discuss future prospects in further characterizing this system, to help narrow down on which evolutionary channel it originated from. Could further timing be used to constrain the presence of a wide companion in a triple (which might have resulted in Kozai-Lidov cycles?). Could phase-resolved spectroscopy in the optical reveal the composition of the donor (and would this be feasible with current optical/IR telescopes given the luminosity of the object and a nominal distance of 4 kpc to M71?). Could a deeper x-ray observation reveal something about the nature of the system which is not apparent in the radio? Exploring questions like these would help make the discovery feel more compelling to the broader community, whereas currently, this is a manuscript that feels to me as though it is targeting the radio pulsar community (and the goal of a journal like Nature is to aim for broader appeal than that).

--Thanks for the suggestions.

In the Methods section, we briefly described the future works on radio timing to measure the orbital period derivative, parallax and proper motion for possible further study on the distance, evolution and origins of this binary. We estimated the possibility of the optical detection to this the companion. The optical identification to the companion would also bring chance to obtain the pulsar mass and the orbital inclination angle. We also mentioned radio observations and the possible simultaneous observations with X-ray satellites to reveal the interaction between the pulsar and its companion.

With all the effort for monitoring this binary, it would provide the chance to understand the relationship between the binary M71E and the globular cluster M71.

Referee #2 (Remarks to the Author):

Nature 2023-03-03727

Pan et al.

"A Binary Pulsar in a 53-minute Orbit"

This is a very interesting, and quite straight-forward, paper about the discovery of the shortest orbital period binary pulsar. The orbital period is 30% shorter than the previous record holder, a gamma-ray black widow system uncovered only a couple years ago.

That alone makes the pulsar quite interesting. But the fact that it also seems to be in a mass range between so-called black-widow and redback pulsars implies a possible evolutionary link between those similar but separate "spider" systems. That does potentially tell us something new about binary evolution.

The one possible caveat to that is the fact that this system *might* be either from, or currently a part of, globular cluster M71. And the stellar dynamics in globular clusters can do crazy things.

Nevertheless, this is a Nature-worthy result, and the paper describes it all quite nicely.

--We thank Dr. Ransom for the positive evaluation of the manuscript and for

providing many helpful comments. We agree with the referee for the caveat about the dynamical channel, which was also pointed out by the first referee. Following the comments from both referees, we have added a discussion on this possibility.

I have only a few relatively minor comments.

-- Nowhere in the paper do the authors discuss whether this pulsar exhibits radio eclipses, which leads me to believe that it does not. Given that "spider" pulsars often eclipse (especially the redbacks), the fact that this one likely does not is surprising. I think this should be explicitly discussed, if only for a couple sentences.

The lack of eclipses (assuming that is correct!) also strengthens the arguments in the paper that suggest that the system is viewed nearly face-on. It is worth mentioning for that point alone.

--We have added some sentences to mention the lack of eclipses both in the main text and in Methods.

If space allowed (even just in the methods) it might be valuable to show either the timing residuals as a function of orbital period (which should be distributed uniformly without eclipses) or a pulse phase vs time greyscale (as shown on a prepfold plot, for instance) for a long observation which covered >1 full orbit. This would especially be useful at the lowest observed frequencies, since eclipses are most apparent there.

--Thanks. Both the orbital phase to residuals and the PRESTO prepfold plots (pulse phase vs time) are now added in the Methods.

-- I would request that the authors include an estimated flux density of the pulsar. That is important for others who might try to observe the system.

--We added the total intensity flux for the pulsar in the Method and the capture of the pulse profile. Please find them in Extended Data Figure 2.

-- Summary: "Observed for tens, ..." Is that a typo? What does it mean?

-- Fig 2 caption: "with the medium" should be "with the median", I think

--Corrected, thanks!

-- Fig 2: This figure seems to be missing J1653-0158, the previous record holder (described in Nieder et al 2020), with PB = 0.052 days. (Note that I'm not as concerned with the other *likely* pulsar mentioned in the first paragraph, FTZ J1406+1222).

--Thanks, we added J1653-0158. While its companion was marked in *psrcat* as a possible white dwarf, we thought that it may not be in such an evolution track and excluded it. We modified the plot and now all the binaries with (possible) white dwarfs or low mass stars are included. ZTF J1406+1222 was not included.

That's all I have. Very nice result!

Scott Ransom

--Thanks again and best wishes!

Sincerely and best wishes!

Peng Jiang, Jinlin Han, Bing Zhang, Zhichen Pan, Jiguang Lu, and Hailiang Chen on behalf of
the coauthors

2023-04-24

Reviewer Reports on the First Revision:

Referees' comments:

Referee #1 (Remarks to the Author):

I believe the authors have thoroughly addressed my concerns, and have produced a stronger manuscript that definitely warrants publication in Nature, and recommend the paper be accepted.

Here are just one or three minor things for the authors to think about (totally optional):

"To narrow down on which evolutionary channel M71E originated from, monitoring observations in multi-bands are necessary. The element abundances of the companion would be estimated by the optical spectra observation with the HST/STIS of G430M/G750M gratings, covering a wavelength range of 3050 to 10100 Å, with a resolving power of ~ 17000 . The optical detection to the companion will also help to determine the companion mass, with which the pulsar mass and orbital inclination angle will be obtained correspondingly"

>It might be worth doing a back-of-the-envelope estimate of how bright you expect the object to be. We know of enough BW pulsars with optical counterparts that the luminosity range is well established for these objects. M71 is at 4 kpc, and the reddening is not too intense, so getting at the optical counterpart in the future seems doable. I think saying something like "this object is likely about XX mag in r band based on the luminosities of other BW companions" would be more useful to discuss than a particular HST setup needed for characterization. The main thing is to give people an idea of whether this object is within the reach of facilities such as HST/JWST (which I think it is!).

>On the subject of the x-ray counterpart--I think it is great that this is now included, and thought maybe it'd be worth mentioning a possible physical origin from the x-rays. In particular, I think one expects roughly the level of flux seen from emission originating in the intra-binary shock. In any case, I think it'd be worth briefly exploring why we expect to see x-ray flux from the system.

>On the He WD channel: so the authors cite literature pointing out that RLO does not occur in He WDs until very short (<10 min) periods--I guess one question I have is whether there is a way for such a system to evolve back out to longer orbital periods via stable mass transfer and then detach into a spider binary (similar to the way AM CVns evolve). The mass ratio is so extreme in these systems that I imagine they would be driven back out to periods much longer than the period minimum near the onset of Roche lobe overflow at <10 mins. Another way of phrasing my question is: are we actually confident that 53 mins is near the minimum period this system reached in its lifetime, or are there evolutionary tracks that might allow an ultracompact x-ray binary that was once down at a period of a few minutes to evolve into the present day object?

Author Rebuttals to First Revision:

Dear referee:

Thanks for these comments. Below are the reply.

Referees' comments:

Referee #1 (Remarks to the Author):

I believe the authors have thoroughly addressed my concerns, and have produced a stronger manuscript that definitely warrants publication in Nature, and recommend the paper be accepted.

Here are just one or three minor things for the authors to think about (totally optional):

"To narrow down on which evolutionary channel M71E originated from, monitoring observations in multi-bands are necessary. The element abundances of the companion would be estimated by the optical spectra observation with the HST/STIS of G430M/G750M gratings, covering a wavelength range of 3050 to 10100 Å, with a resolving power of ~ 17000 . The optical detection to the companion will also help to determine the companion mass, with which the pulsar mass and orbital inclination angle will be obtained correspondingly"

>It might be worth doing a back-of-the-envelope estimate of how bright you expect the object to be. We know of enough BW pulsars with optical counterparts that the luminosity range is well established for these objects. M71 is at 4 kpc, and the reddening is not too intense, so getting at the optical counterpart in the future seems doable. I think saying something like "this object is likely about XX mag in r band based on the luminosities of other BW companions" would be more useful to discuss than a particular HST setup needed for characterization. The main thing is to give people an idea of whether this object is within the reach of facilities such as HST/JWST (which I think it is!).

--Thanks for the suggestion. We estimated the magnitude of the companion according to the luminosity from our model. It is around 30 magnitude and could be possibly observed by HST. We added several sentences in the HST estimation part in Methods.

>On the subject of the x-ray counterpart--I think it is great that this is now included, and thought maybe it'd be worth mentioning a possible physical origin from the x-rays. In particular, I think one expects roughly the level of flux seen from emission originating in the intra-binary shock. In any case, I think it'd be worth briefly exploring why we expect to see x-ray flux from the system.

--Thanks. We simply explained the origin of the x-rays as rotation-powered. Please find the text in the end of the first paragraph on Methods, Multiband Data.

As comparing with the table in Jongsu Lee et al. (2018), the X-ray flux of M71E system is comparable to those of most MSPs without accretion (including the non-accreted MSPs in binaries and isolated MSPs), and is much smaller than those of most spider systems. Thus, it can be expected that the X-ray radiation of M71E is roughly dominated by the rotation-powered component, and the other components due to accretion and shocks are relatively weak. It may imply that the pulsar may stay in the propeller phase and the stellar wind may not sweep over the donor and the accretion stream, which agrees with the the lack of eclipsing.

>On the He WD channel: so the authors cite literature pointing out that RLO does not occur

in He WDs until very short (<10 min) periods--I guess one question I have is whether there is a way for such a system to evolve back out to longer orbital periods via stable mass transfer and then detach into a spider binary (similar to the way AM CVns evolve). The mass ratio is so extreme in these systems that I imagine they would be driven back out to periods much longer than the period minimum near the onset of Roche lobe overflow at <10 mins. Another way of phrasing my question is: are we actually confident that 53 mins is near the minimum period this system reached in its lifetime, or are there evolutionary tracks that might allow an ultracompact x-ray binary that was once down at a period of a few minutes to evolve into the present day object?

--In the plot below, we show the evolution of a NS-He WD binary with a He WD mass of 0.17 solar mass. The orange bar in the plot indicates the location of M71E according to our study. When the orbital period increases to 53.3 min., the WD mass is around 0.01 Msun, significantly smaller than the estimated companion mass of M71E. However, we did not consider the effect of evaporation by pulsar emission, which may widen the binary separation. Therefore, we cannot completely exclude the possibility that M71E is produced from the evolution of NS-He WD binaries. For the same reason, in Figure 2, we added one more line for M71E when the orbital inclination is between 90 and 25.8 degree (random distribution).

Sincerely and best wishes!

Peng Jiang, Jinlin Han, Bing Zhang, Zhichen Pan, Jiguang Lu, and Hailiang Chen on behalf of the coauthors

2023-05-25